# Assessing Social Determinants of Health in a Prenatal and Perinatal Cultural Intervention for American Indians and Alaska Natives

**DOI:** 10.3390/ijerph182111079

**Published:** 2021-10-21

**Authors:** Rosalina James, Martell A. Hesketh, Tia R. Benally, Selisha S. Johnson, Leah R. Tanner, Shelley V. Means

**Affiliations:** 1Urban Indian Health Institute, Seattle, WA 98144, USA; martellh@uihi.org (M.A.H.); tbenally@uw.edu (T.R.B.); selisj@uw.edu (S.S.J.); 2Seattle Indian Health Board, Seattle, WA 98144, USA; 3School of Public Health, University of Washington, Seattle, WA 98195, USA; NAWDIM@gmail.com; 4Native American Women’s Dialogue on Infant Mortality, Seattle, WA 98144, USA; lerahenry@yahoo.com; 5Portland Area Consultant, Healthy Native Babies Project, Seattle, WA 98144, USA

**Keywords:** talking circle, cultural, intervention, cradleboard, wellness, resilience, prenatal, perinatal, native American, indigenous, American Indian, Alaska Native

## Abstract

American Indians and Alaska Natives (AIANs) refer to cultural traditions and values to guide resilient and strength-based practices to address maternal and infant health disparities. Methods: A case study of a culturally-based educational intervention on AIAN maternal and child health. Results: Cultural teachings have successfully been applied in AIAN behavioral interventions using talking circles and cradleboards, but maternal and child health interventions are not well-represented in peer-reviewed literature. Zero publications included interventions centered around cradleboards and safe sleep. Discussion: There is a need for rigorous published research on culturally based interventions and effectiveness on health outcomes for mothers and babies. Conclusions: This paper discusses how a cradleboard educational intervention incorporates national guidelines to address maternal and infant health while mediating social determinants of health.

## 1. Introduction


*“There are structural and ornamental parts to a cradleboard. Each has a purpose, and the parts come together to surround baby in beauty, tradition, and security.”*
—Shelley Means, Ojibwe/Lakota (sixth author)


*“With the cradle board we have a little wood…goes over like a roll bar to protect the baby’s head…But in a traditional one, you use rose stem, a really heavy duty one. And in our culture rose has medicinal properties. And so with my sons’ cradle boards and mine, they all have the rosewood hoop.”*



*“I think for me the strong connection to culture has really given me a strong base. I feel that has allowed me to nurture my children in a way that…In a healthy way, so that they’ve been able to be so successful. So, I feel like their success is coming from the strong base that our ancestors provided for all of us.”*



*“…the connection to all those ladies that came before me made me strong and made me able to deal with not just health stuff, but also living in such a racist society, which also can lead to poor health. And I really truly believe that my connections to traditions have allowed me to be able to flourish, even being away from my homeland and being away from all of those traditions.”*
—Leah Tanner, Nez Perce (fifth author)

American Indians and Alaska Natives (AIANs) experience major disparities in maternal and infant health outcomes. AIANs suffer twice the rate of infant mortality as non-Hispanic whites (NHWs), with AIAN babies under one year 50% more likely to die from complications due to short gestation or low birthweight. AIAN infants are also 2.7 times more likely to die from unintentional injuries [1]. In 2019, AIAN mothers were almost three times more likely to receive late or no prenatal care relative to NHWs [2]. AIAN women are more likely to experience poverty, poor living conditions, lack of health insurance, and sexual or interpersonal violence than white women. All of these affect maternal health outcomes and the environment a woman can provide for her new infant [3,4,5].

An analysis of Neonatal Intensive Care Unit (NICU) admission, infant mortality, and socioeconomic variables using 2016–2018 Natality Data from the National Vital Statistics System (NVSS) within the National Center for Health Statistics (NCHS) revealed significant disparities in maternal and infant health among urban AIAN populations [6]. Data shows a higher incidence of preterm births and low birth weights among AIANs compared to NHWs. AIAN communities suffer from a higher infant mortality rate than NHWs and have a higher proportion of AIAN newborns being admitted to the NICU [6]. AIANs mothers tend to be younger, with a higher proportion qualifying for low-income health insurance and family nutrition programs compared to NHWs. Across all AIAN women who had a birth recorded in NVSS, over half (56.3%) used Medicaid to cover the cost of birth, compared to 27% of NHW women. Approximately 46.9% of AIAN women who gave birth reported receiving supplemental nutrition through a WIC program, as compared to 21.5% of NHWs [6].

Despite well-documented disparities in AIAN health outcomes, there is relatively little in academic publications that document how cultural traditions and values can inform strength-based interventions. The majority of published literature documenting culturally based interventions for AIAN populations focus on talking circle (TC) research; most of these described a positive relationship between culturally based interventions and health. Although there are a small number of papers that examine cultural approaches to health and the history and evolution of swaddling infants in different cultures, there is currently no published literature describing culturally-based interventions with AIANs utilizing traditional cradleboard teachings to improve prenatal and perinatal health.

### Cultural Interventions to Promote Maternal and Child Health

The Native American Women’s Dialog on Infant Mortality (NAWDIM) project was started in 2000 by a group of AIAN women and community activists alarmed by distressing data on disproportionate AIAN infant mortality and maternal child health outcomes [7]. Rates for all other communities were improving, but AIAN and African-American infant mortality were not following the same trend. NAWDIM works with a collaborative of non-profits in the Seattle-King County region to advocate for AIAN issues through policy analysis and monitoring, grassroots mobilization, and public health campaigns. Since 2003, NAWDIM has provided mothers and families with education using the Healthy Native Babies (HNB) project materials tailored to AIAN communities from the National Safe to Sleep campaign, while constructing cradleboards and facilitating TC interventions. This paper explores how NAWDIM cradleboard classes inform AIAN mothers about infant health risk factors while addressing social determinants of health. To do this, we used a focused case study methodology to examine NAWDIM cradleboard classes in context of the research question: How does a grass-roots prenatal and perinatal culturally-adapted intervention promote national maternal and child health outcome goals by addressing social determinants of health?

TCs, in conjunction with creating a safe space for group support, have been used successfully in clinical, behavioral health and health education settings as part of culturally-based interventions [8,9,10,11,12]. Baldwin et al. [13] used TCs to adapt and tailor an existing intervention first developed with the Keetooway-Cherokee Tribe to address AIAN youth alcohol and substance abuse issues, and then tailored the intervention to incorporate distinct cultural differences in three additional tribes. Treatments for AIAN substance-use disorders that are shown to be effective take place in outpatient settings that use ecological family therapy, cognitive–behavioral therapy, and motivational interview interventions [14]. These treatments and the philosophies that frame them provide a non-judgmental structure where participants can reflect and self-explore in an empathetic and supportive environment [15,16]. TCs are a type of experiential learning that has been shown to influence behavior better than other standard modes of educational instruction, and have been used for AIAN health interventions in a number of areas including substance abuse prevention [9,17], nutrition [18], obesity prevention [11], diabetes control and management [16], cancer screening and prevention [10,12,19], and to elicit learning experiences from a transcultural immersion curriculum [20].

Building cradleboards, a traditional Indigenous tool for protecting infants in a safe and warm environment, continue to be a central activity in education interventions with AIAN prenatal and perinatal mothers. Cradleboard construction begins with a strong piece of wood that provides a foundation for baby to sleep. A firm mattress covers the board. Side flaps wrap around the infant and are laced up tight, holding their back to the board and swaddling their arms and legs. These carriers are used to transport the baby and can be strapped onto a parent’s back, freeing the hands to conduct daily activities. A bow arches over the baby’s head as a sort of roll bar to shield from injury and can be draped with a blanket to provide shade from the sun and elements. Bows are adorned with dangling beaded and often colorful objects to distract and entertain the baby while they are still able to see their mother and interact with the world. Cradleboards can also be propped against a wall so a mother can tend to other matters around the house or even sleep, knowing their baby is safe and secure nearby.

Traditionally, cradleboards were made with natural local materials such as deer hide and elegantly beaded or decorated, often with family or clan imagery and designs. Contemporary cradleboards are adorned with fabric and colors that may carry significance or meaning to families (see Figure 1). For example, the fifth author was especially close to her grandmother and chose fabrics with roses or violets, representing a shared love of these flowers and their strong intergenerational bond.

## 2. Materials and Methods

This case study centers around testimony of two NAWDIM community educators that adapted the Healthy Native Babies materials, which incorporate National Safe to Sleep recommendations, to a culturally based intervention. The educators have been delivering classes in Pacific Northwest AIAN communities for two decades.

### 2.1. Sample and Location

As an original hire with NAWDIM in 2001, the fifth author, Leah, developed the idea of using cradleboard classes to support AIAN mothers and reduce the risk of Sudden Infant Death Syndrome (SIDS), and engaged the urban AIAN community for input and feedback. The sixth author, Shelley, was contracted a few years later to facilitate classes as Leah was going onto maternity leave. The program evolved over the years with backing from community and local leaders, as a culturally adapted talking circle approach to support new and expectant mothers and families in tribal and urban Indian communities throughout the greater Puget Sound region and other parts of Washington State. NAWDIM was initially launched with grants that soon ended. Materials and resources to support cradleboard classes have since been funded by small grants and contributions from local tribes and AIAN-focused foundations. Leah and Shelley have continued as co-coordinators for more than 18 years, largely through volunteer time, so limited resources can be used to purchase class materials. Due to these limited resources, demographic information about participants and participant counts in cradleboard classes have not been gathered. NAWDIM has continued to work with AIAN prenatal and perinatal mothers during the pandemic, constructing cradleboards for people when not able to hold in-person events and offering classes when public health recommendations were allowed, using COVID-19 precautions such as safe distancing and masks.

### 2.2. Qualitative Data Collection, Synthesis, and Analysis

We used a collaborative conversational method [21] as a means of gathering knowledge through story. The method is situated within an Indigenous paradigm that is relational, purposeful in intent to decolonize knowledge sharing, and dialogic in that it is informal and flexible. Third and fourth authors conducted two informal conversational interviews with NAWDIM community educators (authors five and six), and a series of follow-up discussions to capture the origin story of NAWDIM. We also collected accounts of the content and practices of the community education intervention, and the cultural significance of centering safe sleep education around cradleboard construction. Using a storytelling guide, these conversations were recorded with the consent of the storytellers. Recordings were then transcribed using Rev.com and summarized around what emerged from the storytellers themselves and all collaborating authors. Analysis involved coding viewed through a lens of Indigenous values and practices; no matter is as intimately well-informed of the relational aspect that occurs between Indigenous researchers and community “subjects”. In accordance with the collaborative conversational method utilized, the subjects are also co-authors; reading, reviewing, and approving all aspects of this report [21]. As authors, the community educators participated in numerous follow-up calls to clarify facts, logistics, and community and cultural details; the subjects edited, and reviewed quotes and content presented in this report for accuracy and clarity.

Transcripts were analyzed and hand-coded by the first author. Quotes were selected to illustrate the program efforts and exemplify elements of a culturally-based intervention that fosters community interconnectedness and maternal and child health and wellness. Draft findings were circulated and discussed among co-authors to ensure the accuracy of qualitative analysis, interpretation, and reporting.

### 2.3. Positionality

All authors are American Indian or First Nations and each brings a lifetime of experience of living and working in tribal and urban Indian communities. Positionality of this case study is inherently biased toward maternal and child health achieved through traditional and cultural teachings and values. The Urban Indian Health Institute is a Tribal Epidemiology Center and a division of the Seattle Indian Health Board. The Urban Indian Health Institute’s mission is Decolonizing Data, For Indigenous People, By Indigenous People. The Indigenous researchers, scholars, and community educators are affiliated with various tribal communities, with unique cultural practices and stories that supported the perspectives presented in this paper. 

### 2.4. Culturally-Based Program Description

NAWDIM created a program utilizing a series of three TCs combined with cradleboard building sessions to build a supportive community for AIAN mothers and infants and deliver Indigenous teachings to improve maternal and child health. They hope these sessions offer an opportunity to help mothers envision a healthy future for their families and descendants. Before each session, the facilitator lays out cradleboard kits that include back boards, pre-cut fabrics in a variety of patterns with pre-sewn flaps, padding for the mattresses, leather for laces, and a bow. The group spends the first half hour learning about the functional parts of the cradleboard and hand-sewing pieces of fabric for their project prior to initiating the first TC. The format of each TC is grounded in traditional Indigenous values and utilizes Indigenous protocol. For example, at the beginning of each TC, the sixth author introduces the focus topic while holding an eagle feather, a practice that cedes the floor to the speaker. When someone finishes talking, the feather is passed around the circle until each has had an opportunity to share. Anyone that is not ready to speak can pass the feather to the person next to them. The direction of the talking piece varies in different tribal cultural practices, so it alternates clockwise or counterclockwise based on the participants in the circle.

The fifth and sixth authors partner with tribes and organizations serving urban AIAN communities to organize these TCs. The authors rely on these partners to recruit new and expectant mothers and provide spaces for the group to meet. Partners typically provide a meal and snacks. A grandmother or Auntie may attend to help with small children or hold the baby while a mother is working on her board. Three TCs are facilitated over the course of a six-hour class. The first includes basic introductions, tribal affiliations, and informal sharing with the group. The women are asked to say a few words about the baby that they’re making the board for and discuss hopes and dreams for baby. The women also discuss a range of related topics such as motherhood, the baby’s personality, an experience of pregnancy or labor, or due dates. People tend to say little during the first TC as they prefer to get started working on the cradleboard. As the day goes on people engage more with each other and the group conversation increases as their project takes shape.

The second TC creates a space for mothers to talk about stressors that they are experiencing, and how she can be supported by others in the group or what resources are available to mitigate stress. The third TC includes education on safe sleep practices as provided through Healthy Native Babies [22], an adaptation of the National Safe to Sleep campaign designed to deliver culturally appropriate risk-reduction messages about sleep-related causes of infant death. Having participants do work with their hands during TC sessions that can sometimes surface difficult topics or experiences is a purposeful element of the intervention. It serves as a calming, methodical distraction for women opening up about stressful or negative life experiences. After the third TC, the mothers have a cradleboard that holds their babies in a secure position and have gained knowledge about safe sleep practices to reduce the risk of a child dying from SIDS.

## 3. Results

Six themes were developed through the analysis including: addressing social determinants of health with cultural interventions, multigenerational connections, developing supportive networks, connections through cultural teachings, and building resilient communities. The final theme mentioned—building resilient communities—was identified as an overarching theme that intersected with all others identified. Each of the other themes identified contributed to this ultimate goal of building resilient communities where AIAN mothers and infants can thrive.

### 3.1. Addressing Social Determinants of Health through Cultural Interventions


*“[One of] the hardest stories where I felt like none of us were equipped to really do more for these women. They had been in residential addiction treatment offered through a diversion program. I don’t even know what form of incarceration. But they’d gotten special permission to come and spend [time in our cradleboard class]…And they were both pregnant and one of them said that her social worker in the system was basically saying, ‘You know that this baby is going to be taken from you. When you have this baby, this baby is going to be taken away from you because your other kids are already in CPS [Child Protection Services].’ And it was so harsh. And so, we’re doing our last talking circle. She’s expressing her gratitude that we provided this day for her. And she’s just in tears because she doesn’t want to go back, and she never gets to find herself in a space like that that’s supportive and safe and nobody there is going to hurt her.”*


Social determinants of health are interwoven throughout this mother’s experience including structural racism built into incarceration systems and child protection services. Social and environmental factors such as poverty, homelessness, or domestic violence that lead young AIANs to intersect with these systems exacerbate existing risk factors. Additionally, these AIAN mothers are trying to navigate a fraught social and logistical landscape with non-existent or inconsistent cultural and community support [23].

Contrast the above with Leah’s opening story. Here, factors are addressed that reduce the risk of SIDS when the baby is positioned to sleep on her back. Babies are not sharing a bed with parents and soft bedding is avoided that could suffocate a child because she is strapped to a board that does not allow her to roll over [24]. This story also illustrates how an expecting mother’s stress level can be reduced when she is in a supportive space and develops tools to keep her baby safe and secure. Reducing maternal stress can also reduce the risk of common stress-related complications that disproportionately affect AIAN populations, such as hypertensive disorders during pregnancy and low birth weight [3,4].

The NAWDIM intervention combines TCs and cradleboard building to deliver culturally-based tools for AIANs to achieve balance in a society that tends to overlook, discount, and marginalize them. Among the tools is the underlying value that Indigenous people care for each other when in need:


*“*
*There were a couple of cousins in the room who of course knew each other. And I think one of the moms and auntie had come to help do the work… a third cousin…had just lost a baby. And we’re in a room of people carrying babies in their womb, and so it was just really, not just heavy, but sort of that deeper spiritual place. It’s like, we couldn’t let these people feel afraid. We couldn’t let them go there. We needed to bring in some smudging and some protection and some prayers. And so the staff there at [tribe] were able to go actually pick up an elder and bring them back to the lodge that we’re working in and helped us all, helped us all tremendously, so that we could kind of move forward in the day and feeling supported in all of that.”*


### 3.2. Multigenerational Connections

Cultural protective factors emerged through conversations that illustrated the interconnectedness that tie families and community together across generations:


*“[M]y ancestors were in the Nez Perce War of 1877. And my great, great grandma, went through that whole thing pregnant, and saw her mother killed at the Battle at Big Hole, in Montana, but escaped up into Canada with her husband and her father where she gave birth to my great-grandmother…And every time I have my cradleboard, I always thought of her…I can’t be weak. She wasn’t. And that’s the women that I come from. I need to be able to carry that and then hopefully pass it on to granddaughters. I have my fingers crossed because I have five sons. One of these days, I hope to have granddaughters…That’s where I come from, why it’s important to me to keep our cradleboard traditions alive.”*


The educational approach used in NAWDIM cradleboard classes expose AIAN mothers and families to cultural values and traditions that build strong connections to self, to their child, and across generations. “*It was a mom and her daughter…she just came along with her mom and her grandma who were making [a cradleboard] for another family member…during the talking circles…She had her story about the impact on the family, and her prayers for this new baby that was on its way…Then I saw this young woman maybe five years later, and she said that she actually had gotten pregnant…She made a board for her first child, and then maybe three or four years later came back and made another one*.”

### 3.3. Connections through Cultural Teachings

Many AIANs who reside in urban settings are far from their community where cultural knowledge is typically shared. Cultural teachings can bridge a divide for AIANs to create that base knowledge of the rich people and cultures that they are connected to so that they can take steps to prepare for healthy and holistic futures for their own children: “*Whenever I travel, I always carry something from Idaho wherever I go… And it always was a comfort to me to have a piece of Idaho, of home with me. And I imagine that you all can probably relate to that because even though I’ve been over here for so long, Idaho’s always going to be my homeland. And it’s always the place that I… When I dream about feeling safe, it’s always there at my grandma’s house*.”

Due to historic trauma and policies that have disenfranchised AIAN populations, many have struggled to understand their own identity without the direct experience of learning traditions from the community or family or knowing their home landscapes. The cradleboard intervention provides tools and a space for mothers to learn and practice traditional teachings. “*A lady who hadn’t been in her home community, I don’t think ever, [talked] about how healing it felt to be able to create something that then she could pass down to her family. And it’s something specifically Indigenous. That was just a benefit that I didn’t see coming at all. Because again, I was raised pretty traditionally and had access to all of that and realize even now with folks on the rez, sometimes they don’t have access to our traditional ways too*.”

Bolstered by culture, tradition, and interconnectedness, these protective factors contribute to self-efficacy through traditional teachings and practices. “*I can’t participate in all the different ceremonies that happen throughout the year…But I still hear the songs in my dreams and when I’m feeling particularly vulnerable my family comes to me in my dreams, and I hear the songs and it gives me strength… And for me, it’s not just the physical health, but mental and spiritual health too.*”

### 3.4. Developing Supportive Networks

A critical element of the NAWDIM intervention is engagement of AIANs through a community network of support. “…a lot of our [NAWDIM] members [partners] are direct service providers from community-based organizations. There’s community building that we foster at these different levels. One is among the service providers, the doulas, the home visitors…we’ve really built this culture of making space for whoever needs to be in a class from whichever agency…There’s no strict ownership and programmatic territory. It is precisely this network that is activated when needed: There’s a woman who was working with one of our members [Home Visitors]. She ended up losing the baby. [The woman] had made a cradleboard with us years ago…We were able to support the Home Visitor…just by being available to figure out some meal cards … and to run a GoFundMe to help them cover the unexpected funeral expenses.” This coordinated effort allowed NAWDIM to work with the Home Visitor that was already providing support without overwhelming the grieving family.

The community also comes together to celebrate new babies at the annual Seafair Powwow. “After the grand entry and before the tiny tots, we do an honoring…as people are arriving, we go around and just make sure everybody we see that has a baby knows they can come out for the giveaway.” Partner organizations contribute items such as “little onesies with the powwow logo…lavender sachets, little socks, and hats, and baby things…Then people have program materials, and we just do a giveaway…It’s people who are already in our circles because they utilize services from our community organizations, and/or they’ve been through cradleboard class…We have some beautiful pictures of competitive dancers in their regalia, holding a tiny baby, and stepping into the circle. They get a welcome from our local community as well…We put a blanket out by where the announcer stand is, and put all of our things that we’re going to give away [on it]. There’s a round dance…with drummers and an honor song.”

## 4. Discussion

The NAWDIM cradleboard program represents a culturally-based intervention that operates on Indigenous values, and relies heavily on storytelling and oral traditions as documentation of improved health and wellness. The cradleboard program at NAWDIM was developed by community members in a direct response to community need. Due to their close connection and engagement with the community, NAWDIM recognized that an effective program to address the social determinants of maternal and child health for AIAN communities needed to be based in traditional cultural knowledge. By incorporating TCs into a cradleboard education intervention, the program promotes healing in balance with components of the medicine wheel—emotional, physical, spiritual, and mental [25]. By examining this case study through an Indigenous framework, we see how the NAWDIN cradleboard program addressed different components of the medicine wheel to achieve balance and, therefore, health for mother and infant. According to Patchell’s Circular Model of Cultural Tailoring, an illness can be described as a disruption of the flow of energy. Healing requires a circular nature of life with an “ordered flow and movement in relationships and within the interactions among those relationships [25].” When considering Patchell’s model in this case study, it becomes clear why multiple themes centered around relationships emerged during these conversations. Additionally, program participants self-identify as AIAN as opposed to providing proof of tribal enrollment. This helps create a non-judgmental environment where people feel safe to learn, heal, and engage in infant safe sleep best practices. Requiring proof of tribal enrollment just perpetuates ongoing efforts rooted in colonial practices designed to assimilate tribes and AIANs.

The HNB project has provided training and occasional stipends for AIAN communities to tailor HNB materials for more than a decade in five Indian Health Service Areas with high infant mortality [26]. The National Institute of Child Health and Human Development, who funds the HNB project, announced that this program will be sunsetting by May of 2022 to focus on funding research. Originally developed through a series of convenings in tribal and urban Indian communities, HNB has created key safe sleep messages translated into Indigenous languages, a toolkit, and training—like cradleboard classes to reduce the risk of SIDS among AIANs.

### 4.1. Strengths and Limitations

Indigenous peoples have always been researchers, engineers, scientists, and interventionists guided by holistic views of the world, views that are not well-accepted or understood by western concepts of medicine and public health. “*But we’ve always looked at it [holistic health] that way as native people…it’s so interesting to me how we come from these different places, but we have so many similar, similar beliefs, which is such a strength to me*.” Addressing complex issues such as the social determinants of health requires a holistic approach that accounts for historical and current influences on the health of communities. By including program facilitators as authors through a collaborative method, this paper accounts for those complexities to provide a more nuanced and accurate interpretation of the community impacts of the NAWDIM cradleboard program.

Given the propensity for data discrepancies representing AIANs, the socioeconomic and health disparities described in this paper informed by NVSS data may well be underestimated. Public health data collected and reported by local, state, and federal agencies consistently undercounts or misrepresents the AIAN population, often collapsing this data into groups like “other” where more than one race or ethnicity is marked. In addition to being a relatively small population, practices like this coupled with racial misclassification contribute to underrepresenting the true extent of health disparities experienced by AIAN mothers and infants [27,28]. Additionally, no participant information such as income status, insurance information, or education level has been recorded for participants in the NAWDIM cradleboard program. This is because there are no paid staff or consultants to manage this data collection. Due to being chronically under-funded and under-resourced, the program has not had the opportunity to systematically collect participant demographics or longitudinal health outcomes that we can report at this time.

### 4.2. Implications for AIAN Maternal and Child Health

This case study identifies a significant gap in the literature on AIAN culturally based interventions, with no published literature addressing cradleboards and safe sleep education at the time of writing. We also demonstrate the need for culturally-based and community-driven interventions such as the NAWDIM cradleboard classes to be funded at consistent and appropriate levels. Additionally, rigorous research assessing the efficacy and effectiveness of Indigenous culturally-based safe sleep, maternal support, and holistic health interventions is needed, particularly by Indigenous researchers using Indigenous methodologies. Although additional research into how cradleboard class interventions and safe sleep campaigns can reduce negative AIAN maternal and child health outcomes is necessary, researchers must always be mindful that “*Research won’t mean anything if we don’t have connection to community*”.

## 5. Conclusions

American Indians and Alaska Natives experience maternal and infant health disparities while consistently referring to cultural traditions and values to guide resilience and culturally-based interventions. The evidence for how well these interventions work to improve AIAN maternal and child health is not well-represented in peer-reviewed literature. There is a need for rigorous published research on culturally based interventions and effectiveness on health outcomes for mothers and babies. However, research should be complementary to, rather than in place of, funding cultural community-level programming. This paper reports on how a grass-roots cradleboard educational intervention incorporates national guidelines to address AIAN maternal child health recommendations while mediating social determinants of health.

## Figures and Tables

**Figure 1 ijerph-18-11079-f001:**
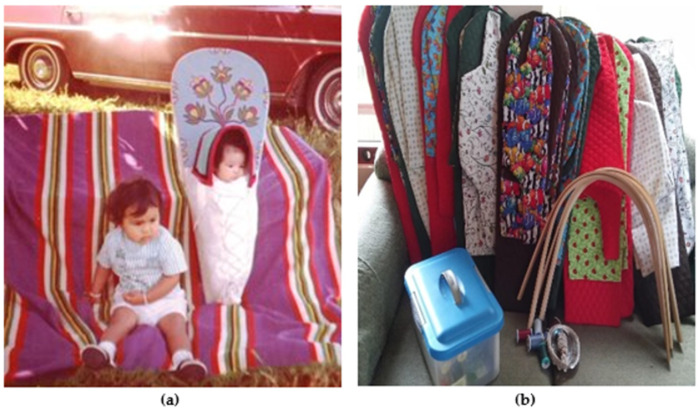
Cradleboards are a form of traditional engineering for carrying protecting babies (**a**) panel a includes the fifth author at one year of age next to her sister in a traditional Nez Perce cradleboard; (**b**) panel b showcases NAWDIM contemporary cradleboard class materials.

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
