# Peer review of "Assessing Social Determinants of Health in a Prenatal and Perinatal Cultural Intervention for American Indians and Alaska Natives"

_ijerph, 2021, doi:10.3390/ijerph182111079_

Round 1

Reviewer 1 Report

Correct: in line 53 is repeated "Culture-based interventions" 

A solidarity and intergenerational project has been developed. Cultural interventions are intimately related to Citizen Science; this methodology is of high interest both by the scientific community and by numerous political and social institutions worldwide. It is a tool for collaboration between different groups, through which citizens are directly involved in the research process. and actively contribute, either with their intellectual effort, with knowledge of their environment or by providing their own tools and resources.

Congratulations on your work and best of luck. I hope that your efforts will be rewarded and that you will not withdraw funding for this wonderful project.

Author Response

Response to reviewer comments: Author responses to each comment are in bold type

Review 1

 Correct: in line 53 is repeated "Culture-based interventions" 

Removed duplicate phrase

A solidarity and intergenerational project has been developed. Cultural interventions are intimately related to Citizen Science; this methodology is of high interest both by the scientific community and by numerous political and social institutions worldwide. It is a tool for collaboration between different groups, through which citizens are directly involved in the research process. and actively contribute, either with their intellectual effort, with knowledge of their environment or by providing their own tools and resources.

Congratulations on your work and best of luck. I hope that your efforts will be rewarded and that you will not withdraw funding for this wonderful project.

Reviewer 2 Report

Thank you for the opportunity to read about the Native American Women’s Dialog on Infant Mortality (NAWDIM) project. This is a very interesting paper and there are several things I would like to complement the authors on:

The inclusion of participants as authors - this is very eloquently justified in the paper, and should be encouraged more in research as we move towards more community embedded research methods. 

The positionality and limitations section is particularly helpful, stating rationally the aims of the paper, the position from which it is written and the limitations which are realistic. 

The description of the intervention is comprehensive and includes more than just a description of the process - I felt I understood the approach, the cultural reasons for the intervention and the mechanisms by which change might be achieved. We need more of this in complex intervention research. 

The only issue I have with the results from the interviews are that they are a little short and I would have liked to hear more. I am curious as to whether the paper might benefit from either leaving out the literature review (this could be a separate paper), or including it in the introduction as background to the intervention. Similarly, the data analysis from the National Vital Statistics System could also be usefully summarised in the introduction, leaving space for the results section to be slightly longer. I might also suggest some headings for each aspect of the results, just to help readers find their favourite bits more easily when re-reading the paper, as I am sure many will. 

I hope these suggestions are useful to the authors, and thank you once again for this very interesting paper on a very important topic. 

Author Response

Response to reviewer comments: Author responses to each comment are in bold type

Review 2

2.1) Thank you for the opportunity to read about the Native American Women’s Dialog on Infant Mortality (NAWDIM) project. This is a very interesting paper and there are several things I would like to complement the authors on:

The inclusion of participants as authors - this is very eloquently justified in the paper, and should be encouraged more in research as we move towards more community embedded research methods. 

The positionality and limitations section is particularly helpful, stating rationally the aims of the paper, the position from which it is written and the limitations which are realistic. 

The description of the intervention is comprehensive and includes more than just a description of the process - I felt I understood the approach, the cultural reasons for the intervention and the mechanisms by which change might be achieved. We need more of this in complex intervention research. 

The authors appreciate these comments and opportunity to publish our work

2.2) The only issue I have with the results from the interviews are that they are a little short and I would have liked to hear more.

Thank you for this suggestion. We have included additional stories from the interviews that illustrate further how the intervention directly addresses social determinants of health in the lives of Native mothers and families through building community and support networks. These examples can be found on pages 6 to 8

2.3) I am curious as to whether the paper might benefit from either leaving out the literature review (this could be a separate paper), or including it in the introduction as background to the intervention. Similarly, the data analysis from the National Vital Statistics System could also be usefully summarised in the introduction, leaving space for the results section to be slightly longer.

Recognizing that articles for this special issue do not have a word limit, and in response to Review 3 comments, we decided to add subheadings to clarify the three types of data sources and keep these findings in the Results section. We believe that the gaps in peer reviewed published literature show that there is a dearth of documented research on culturally-integrated maternal and child interventions and the dataset analysis emphasizes disparities between AIANs and NHWs and a need for effective interventions.

2.4) I might also suggest some headings for each aspect of the results, just to help readers find their favourite bits more easily when re-reading the paper, as I am sure many will. 

Subheadings have been included in the Results section.

I hope these suggestions are useful to the authors, and thank you once again for this very interesting paper on a very important topic. 

Reviewer 3 Report

Dear authors,

Thank you for the paper submitted covering an interesting topic. While the article provides a very useful exploration of the culture-based interventions in order to reduce maternal and infant health disparities, there are some aspects of the article that should be improved. Furthermore, there are some issues about the references format that are totally wrong and inexcusable in a submission to a peer-reviewed journal. For instance, references must be numbered in order of appearance in the text, but this circumstance is not followed by authors in the paragraph contained between line 53-68. This circumstance denotes a lack of seriousness in scientific writing. Also there is no thread that allows to understand the aim of the study. It is difficult to understand how the study has been carried out because the authors first focus on the conversational interviews with the community educators and after on the Talking circles with the pregnant women.

There are a few specific areas that also need further clarification to work on.

Abstract:

Line 18: Non-hispanic whites: The acronym should be introduced here, because it is mentioned later.

Introduction:

This section is not well structured, there is a baseline (disparities in maternal and infant health outcomes) but afterwards, there is no thread running through this section. 

Lines 27-42: I suggest to remove these propositions from the introduction section, because they don´t provide scientific background and rationale. The readers don´t know whom they belong to  when placed at the beginning of the paper.

Line 48: Non-hispanic whites: The acronym shot

Line 53: Culture-based intervention is mentioned twice.

Lines 52-68: Talking circle and cultural interventions are described in this paragraph. This paragraph is not well introduced and it refers to methodology, so it should be placed in Materials and Methods section. I have to mention again the inexcusable error regarding the references.

Lines 69-75: Now, the cradleboards are mentioned. This paragraph has no connection to the previous one. I can’t understand why they are included and the relationship with the previous information.

Lines 85-87: The objective/aim of this paper is confusing. The research question is well formulated in the following section:

“How does a grass-roots prenatal and perinatal culturally-adapted intervention address national maternal and child health outcome goals by addressing social determinants of health?”

I suggest to place it in the introduction section.

Materials and Methods:

Lines 92-97: There are three sources of data instead of two.

Lines 99-110:  There are no specifications of the sample (expectant mothers and families…) that are going to participate in the Talking circles. There are no sociodemographic data included in this section. There are only two subjects interviewed that are also authors of the paper. So, they were aware of the aim of the paper and manipulated the data (lines 125-126: “reading, reviewing and  approving all aspects of this report”). This constitutes a bias that invalidate the study.

Line 112-129: the data analysis is not explained 

Line 140: Secondary data analysis is mentioned twice.

Line 141: NICU: I suggest to use neonatal intensive care unit instead of the acronym because it has not been introduced before. 

Line 141-142:  A reference must be introduced for : 2016-2018 Natality Data from the National 141 Vital Statistics System (NVSS) within the National Center for Health Statistics (NCHS) 

Lines 169-185: This paragraph does not provide any information for the replication of the intervention. The information is redundant, because it has been already mentioned in the introduction.

Lines 187-189: There are two pictures labelled with “a” and “b” but the caption is misleading because it mention “a” and “c”.

Line 200: Talking circles (TC) :The acronym should be introduced before.

Results:

There is no qualitative methodology in the results. The results are not conclusive, and it is  included a subjective appreciation of the authors. This sections begins with a proposition followed by two paragraphs with no relationship with any result, they could belong to the discussion. Even the information mentioned has its references to other papers. 

Lines 300-303: The figure must be improved

I suggest to the authors restructure the Result section in three subsections:

-Qualitative results 

-Literature review

-Analysis of a national dataset. (If the authors presents these data, sociodemographic factors are requested ).

Discussion: 

This section is poorly presented with only two paragraphs in which the results are not discussed. What it is mentioned is the background and the future perspectives of the Healthy Native Babies project.

Line 339: Healthy Native Babies project (HNB) 

Conclusions:

The conclusions mentioned in this sections are not sustained by any information registered in the paper.

References:

Please check the journal´s recommendations, all the references don´t follow the ACS style guide.

Author Response

Response to reviewer comments: Author responses to each comment are in bold type

Review 3

3.1) Thank you for the paper submitted covering an interesting topic. While the article provides a very useful exploration of the culture-based interventions in order to reduce maternal and infant health disparities, there are some aspects of the article that should be improved. Furthermore, there are some issues about the references format that are totally wrong and inexcusable in a submission to a peer-reviewed journal. For instance, references must be numbered in order of appearance in the text, but this circumstance is not followed by authors in the paragraph contained between line 53-68. This circumstance denotes a lack of seriousness in scientific writing.

The reference format and misnumbered references have been corrected.

3.2) Abstract:

Line 18: Non-hispanic whites: The acronym should be introduced here, because it is mentioned later.

Editors: Is it the Journal’s practice to introduce acronyms in the abstract? I typically spell out everything in abstract and introduce any acronyms when first mentioned in the main text of the narrative because people often read the narrative separate from abstract. I’m happy to comply with either practice.

3.3) Also there is no thread that allows to understand the aim of the study. It is difficult to understand how the study has been carried out because the authors first focus on the conversational interviews with the community educators and after on the Talking circles with the pregnant women.

Introduction: This section is not well structured, there is a baseline (disparities in maternal and infant health outcomes) but afterwards, there is no thread running through this section. 

 The Introduction section has been restructured with the following thread: AIANs suffer disparities in maternal and child health àtalking circles have been successfully used in cultural interventions à talking circles in combination with cradleboard construction may be a culturally-based intervention for AIAN safe sleep and mothers and families. This thread ties with the aim of the case study, to examine the research question of how a grass roots intervention promotes national maternal and child health outcome goals by addressing social determinants of health

3.4) Lines 27-42: I suggest to remove these propositions from the introduction section, because they don´t provide scientific background and rationale. The readers don´t know whom they belong to  when placed at the beginning of the paper.

We appreciate the reviewer’s suggestion. However, the methodology utilized in this case study purposefully engages culturally rigorous practices. Therefore, story is central to how data was collected and opening with story in the introduction sets the foundation for a “thread” that runs throughout the piece. For example, Leah’s comment about how her ties to past generations of mothers and their cradleboard practices have strengthened her self-efficacy to face adversity and racism, exemplify ways Indigenous culture and teachings build self-efficacy. The authors have bolstered that foundation with references and analysis directly tied to opening story that can be found in the introduction, qualitative results section, and discussion.

3.6) Line 48: Non-hispanic whites: The acronym shot

The acronym NHWs has been included here.

3.7) Line 53: Culture-based intervention is mentioned twice.

This duplication of the phrase has been corrected.

3.8) Lines 52-68: Talking circle and cultural interventions are described in this paragraph. This paragraph is not well introduced and it refers to methodology, so it should be placed in Materials and Methods section. I have to mention again the inexcusable error regarding the references.

 A transition sentence connecting AIAN infant and maternal disparities to establish the published evidence of scientifically rigorous intervention methods has been added. This paragraph provides readers with background on how culturally informed practices have been studied and shown to produce improved health and behavioral health outcomes.

3.9) Lines 69-75: Now, the cradleboards are mentioned. This paragraph has no connection to the previous one. I can’t understand why they are included and the relationship with the previous information.

The cradleboard description and cultural significance has been moved to and further developed in the introduction and linked with culturally-based interventions such as talking circles.

3.10)Lines 85-87: The objective/aim of this paper is confusing. The research question is well formulated in the following section:

“How does a grass-roots prenatal and perinatal culturally-adapted intervention address national maternal and child health outcome goals by addressing social determinants of health?”

I suggest to place it in the introduction section.

The research question has been moved to Introduction section

 3.11) Materials and Methods:

 Lines 92-97: There are three sources of data instead of two.

 This has been corrected

3.12) Lines 99-110:  There are no specifications of the sample (expectant mothers and families…) that are going to participate in the Talking circles.

A key point of the case study is that programs such as NAWDIM are underfunded and understudied. The program has not had formal assessment or evaluation of the population that it serves as resources tend to be put towards program supplies and basic costs. The authors have added more context to Discussion section (lines 424-437)

3.13) There are no sociodemographic data included in this section.

Sociodemographic data has been added including age, insurance coverage, and WIC program status.

3.14) There are only two subjects interviewed that are also authors of the paper. So, they were aware of the aim of the paper and manipulated the data (lines 125-126: “reading, reviewing and  approving all aspects of this report”). This constitutes a bias that invalidate the study.

Thank you for this observation. However, the Conversational Method (Kovach, 2010) used for this case study does not see the method of data collection as the determining characteristic so much as the interplay (or relationship) between the method and Indigenous paradigm. How activities, such as interviews, are carried out is important. And these research activities must reflect protocols that are done in a good way, which involves Indigenous researchers collecting and interpreting the data through a community and cultural lens. Finally, the authors were clear and transparent about any biases related to the study design so readers can assess the findings and conclusions with limitations in mind.

3.15) Line 112-129: the data analysis is not explained 

Data analysis description has been added

3.16) Line 140: Secondary data analysis is mentioned twice.

This has been corrected

3.17) Line 141: NICU: I suggest to use neonatal intensive care unit instead of the acronym because it has not been introduced before. 

NICU has been spelled out

3.18) Line 141-142:  A reference must be introduced for : 2016-2018 Natality Data from the National 141 Vital Statistics System (NVSS) within the National Center for Health Statistics (NCHS) 

 A reference to the dataset has been added.

3.19) Lines 169-185: This paragraph does not provide any information for the replication of the intervention. The information is redundant, because it has been already mentioned in the introduction.

 This suggestion is appreciated. The cradleboard description has been moved to the introduction.

3.20) Lines 187-189: There are two pictures labelled with “a” and “b” but the caption is misleading because it mention “a” and “c”.

 The labelling has been corrected.

3.21) Line 200: Talking circles (TC) :The acronym should be introduced before.

 The acronym has been introduced at first mention

3.22) Results: There is no qualitative methodology in the results. The results are not conclusive, and it is  included a subjective appreciation of the authors. This sections begins with a proposition followed by two paragraphs with no relationship with any result, they could belong to the discussion. Even the information mentioned has its references to other papers. 

 Qualitative analysis section subheading has been added to clearly identify findings in the results section

3.23) Lines 300-303: The figure must be improved

 This has been corrected. The journal article template and figure table was difficult to edit. A separate figure will be uploaded in case the one we have inserted still doesn’t look right

3.24) I suggest to the authors restructure the Result section in three subsections:

-Qualitative results 

-Literature review

-Analysis of a national dataset. (If the authors presents these data, sociodemographic factors are requested ).

These subheadings have been added to the Results section and sociodemographic factors included in report

3.25) Discussion: 

This section is poorly presented with only two paragraphs in which the results are not discussed. What it is mentioned is the background and the future perspectives of the Healthy Native Babies project.

 The Discussion section has been revised to include discussion of literature review, dataset analysis, and qualitative findings.

3.26) Line 339: Healthy Native Babies project (HNB) 

 The acronym has been added

3.27) Conclusions:

The conclusions mentioned in this sections are not sustained by any information registered in the paper.

With revisions to Discussion and the additions of findings, the conclusions align with the arc presented throughout the article. For example, Qualitative Results section analyze the data from interviews and identify clear examples of how cradleboard interventions promote safe sleep and maternal and child health outcomes, and then are tied to addressing social determinants of health.

 3.28) References:

Please check the journal´s recommendations, all the references don´t follow the ACS style guide.

References and citations have been corrected

Round 2

Reviewer 3 Report

Thank you for taking the time to address comments on the manuscript entitled:  “Assessing Social Determinants of Health in a Prenatal and Perinatal Cultural Intervention for American Indians and Alaska Natives ”. 

The manuscript has been much improved but it is difficult to read and to understand.  Also there is a lack of thread and the authors jump from one thing to another. 

Although my comments are related to form issues, there is still much work to do to make this paper worth to be published in a JCR journal. In my opinion, the study is very ambitious and maybe, exploring the talking circle strategy is enough.

Authors may improve their scientific writing to build a proper paper structure. 

Introduction:

This section is not well organized , and it doesn’t give the background required.

Lines 54-67: Although the fact of introducing this paragraph has the the purpose of providing scientific background, all this information and the way it is presented makes the paper more difficult to read. It should be described briefly. When I suggested to introduce the sociodemographic data in the paper, I was referring to the data of the study carried out. In lines 401-403 you introduce that you did not collect this information because there were no paid staff, although the study counted on funders. This is an important limitation that should be placed in section 2.5 .

Lines 68-92: These data are related to results. Please place them in the correct section.

Line 104:Use TC (please homogenize in the whole text)

Line 106: Baldwin et. al. Please remove the full stop after et

Material and Methods

Line 162-178: How many participants took part in the study? And the sociodemographic data? This paragraph describes the history of the implementation of the strategies described and all the inconveniences to which the authors faced up. 

Lines 166-167: “The fifth author was contracted a few years later to facilitate classes as Leah was  going onto maternity leave.” Useless.

Line 173: Leah and Shelley. It should be replaced by: The main researchers

Line 233:  Where is reference 21?

Results:

Lines 279-281:  I was surprised when I have read this sentence and furthermore with the references that are not related to this affirmation, specially reference 25. 

Discussion: 

References 

Please check the journal´s recommendations, some of the references do not follow the ACS style guide. (Abbreviated Journal Name in cursiva). 

Some of the links are wrong.

Author Response

Responses to reviewer comments are indicated by bolded lettering. 

Introduction:

This section is not well organized , and it doesn’t give the background required.

This section has been re-organized to improve flow and more succinctly summarize existing literature and data on maternal and child health disparities among AIAN communities.

Lines 54-67: Although the fact of introducing this paragraph has the purpose of providing scientific background, all this information and the way it is presented makes the paper more difficult to read. It should be described briefly. When I suggested to introduce the sociodemographic data in the paper, I was referring to the data of the study carried out. In lines 401-403 you introduce that you did not collect this information because there were no paid staff, although the study counted on funders. This is an important limitation that should be placed in section 2.5 .

 Section was significantly edited down to focus on briefly summarizing data as part of the background on AIAN maternal and infant health disparities. The limitation of a lack of resources to collect demographic information is discussed in a strengths and limitations section added into the discussion.

Lines 68-92: These data are related to results. Please place them in the correct section.

Section was significantly edited down to focus on summarizing data as part of the background on AIAN maternal and infant health disparities.

Line 104:Use TC (please homogenize in the whole text)

 References to TCs have been homogenized.

Line 106: Baldwin et. al. Please remove the full stop after et

 Full stop removed.

Material and Methods

Line 162-178: How many participants took part in the study? And the sociodemographic data? This paragraph describes the history of the implementation of the strategies described and all the inconveniences to which the authors faced up. 

Sentences were added to note how counts and sociodemographic data were not able to be gathered due to lack of funding for these community interventions.

Lines 166-167: “The fifth author was contracted a few years later to facilitate classes as Leah was going onto maternity leave.” Useless.

Noted. Authors chose to leave this in to highlight the collaborative roots of NAWDIN that form the foundation of the supportive community that is discussed in the findings.

Line 173: Leah and Shelley. It should be replaced by: The main researchers

Leah and Shelley are the co-facilitators of this program. Calling them the primary researchers could confuse readers. We use their names in the text of the paper to honor the work they have done in our community.

Line 233:  Where is reference 21?

Errors in reference order have been fixed.

Results:

Lines 279-281:  I was surprised when I have read this sentence and furthermore with the references that are not related to this affirmation, specially reference 25. 

 Sentence was clarified and references were updated to support information cited in the sentence.

Discussion: 

References:

Please check the journal´s recommendations, some of the references do not follow the ACS style guide. (Abbreviated Journal Name in cursiva). 

References were checked and edited to be in accordance with the MDPI journal style guide.

Some of the links are wrong.

All links have been checked and updated if necessary.